# PromptTTS 2: Describing and Generating Voices with Text Prompt

**Yichong Leng**[*†]**, Zhifang Guo**[*]**, Kai Shen, Zeqian Ju, Xu Tan, Yanqing Liu, Yufei Liu**
**Dongchao Yang, Leying Zhang, Kaitao Song, Lei He, Xiang-Yang Li, Sheng Zhao**
**Tao Qin, Jiang Bian**

MOE-Microsoft Key Laboratory of Multimedia Computing and Communication, University of
Science and Technology of China
Microsoft

## Abstract

Speech conveys more information than text, as the same word can be uttered in various voices to convey diverse information. Compared to traditional text-to-speech (TTS) methods relying on speech prompts (reference speech) for voice variability, using text prompts (descriptions) is more user-friendly since speech prompts can be hard to find or may not exist at all. TTS approaches based on the text prompt face two main challenges: 1) the one-to-many problem, where not all details about voice variability can be described in the text prompt, and 2) the limited availability of text prompt datasets, where vendors and large cost of data labeling are required to write text prompts for speech. In this work, we introduce PromptTTS 2 to address these challenges with a variation network to provide variability information of voice not captured by text prompts, and a prompt generation pipeline to utilize the large language models (LLM) to compose high quality text prompts. Specifically, the variation network predicts the representation extracted from the reference speech (which contains full information about voice variability) based on the text prompt representation. For the prompt generation pipeline, it generates text prompts for speech with a speech language understanding model to recognize voice attributes (e.g., gender, speed) from speech and a large language model to formulate text prompts based on the recognition results. Experiments on a large-scale (44K hours) speech dataset demonstrate that compared to the previous works, PromptTTS 2 generates voices more consistent with text prompts and supports the sampling of diverse voice variability, thereby offering users more choices on voice generation. Additionally, the prompt generation pipeline produces high-quality text prompts, eliminating the large labeling cost. The demo page of PromptTTS 2 is available[1].

## 1 Introduction

In recent years, there have been significant advancements in text-to-speech (TTS) systems (Wang et al., 2017; Popov et al., 2021b; Chen et al., 2022b), which have resulted in enhanced intelligibility and naturalness of synthesized speech (Tan et al., 2021). Some TTS systems have achieved a level of quality comparable to that of single-speaker recording (Tan et al., 2022), and large-scale TTS systems have been developed for multi-speaker scenarios (Wang et al., 2023; Le et al., 2023). Despite these improvements, modeling voice variability remains a challenge, as the same word can be delivered in various ways such as emotion and tone to convey different information. Conventional TTS methods often rely on speaker information (e.g., speaker ID) (Gibiansky et al., 2017) or speech prompts (reference speech) (Casanova et al., 2022) to model the voice variability, which are not user-friendly, as the speaker ID is pre-defined and the suitable speech prompt is hard to find or even does not exist (in voice creation scenario). Given that natural language is a convenient interface for users to express

---

[*]Equal contribution. lyc123go@mail.ustc.edu.cn; zhifangguo9@gmail.com
[†]This work was conducted at Microsoft. Corresponding author: Xu Tan, xuta@microsoft.com

[1]https://speechresearch.github.io/prompttts2

their intentions on voice generation, a more promising direction for modeling voice variability is to employ text prompts (Guo et al., 2023; Ramesh et al., 2022; Brown et al., 2020b) that describe voice characteristics. This approach enables easy voice creation through text prompt writing.

In general, TTS systems based on text prompts are trained with a text prompt dataset, consisting of speech and its corresponding text prompt. Voice is generated by model conditioned on the text content to be synthesized and the text prompt describing the variability or style of the voice. Two primary challenges persist in text prompt TTS systems:

- **One-to-Many Challenge**: Speech contains voice variability in detail, making it impossible for text prompts to fully capture all characteristics in speech. So different speech samples can correspond to the same text prompt [2]. This one-to-many mapping increases the difficulty of TTS model training, leading to over-fitting or mode collapse. To the best of our knowledge, no mechanisms have been specifically designed to mitigate the one-to-many issue in TTS systems based on text prompts.
- **Data-Scale Challenge**: Dataset of text prompts describing the voice is hard to construct since the text prompt is rare on the internet. So venders are engaged to compose text prompts, which is both costly and laborious. Consequently, the text prompt datasets tend to be relatively small (approximately 20K sentences) (Guo et al., 2023) or not openly accessible (Yang et al., 2023), posing an obstacle for the future research on text prompt based TTS systems.

To address the aforementioned challenges, in our work, we introduce PromptTTS 2 that proposes a variation network to model the voice variability information of speech not captured by the text prompts and utilizes a prompt generation pipeline to generate high-quality text prompts:

For the one-to-many challenge, we propose a variation network to predict the missing information of voice variability from the text prompt. The variation network is trained with the help of a reference speech, which is regarded to contain all information about voice variability (Wang et al., 2023; Shen et al., 2023). Generally, the TTS model in PromptTTS 2 consists of a text prompt encoder for text prompts, a reference speech encoder for reference speech, and a TTS module to synthesize speech based on the representations extracted by text prompt encoder and reference speech encoder. Variation network is trained to predict the reference representation from reference speech encoder based on the prompt representation from text prompt encoder [3]. By employing the diffusion model (Song et al., 2020) in the variation network, we can sample different information about voice variability from Gaussian noise conditioned on text prompts to control the characteristics of synthesized speech, and thus offering users greater flexibility in generating voices and alleviating the one-to-many issue.

For the data-scale challenge, we propose a pipeline to automatically generate text prompts for speech with a speech language understanding (SLU) model to recognize voice attributes (e.g., gender, speed) from speech and a large language model (LLM) to compose text prompts based on the recognition results. Specifically, we employ a SLU model to describe the voice from many attributes (e.g., emotion, gender) by recognizing the attribute values for each speech sample within a speech dataset. Subsequently, sentences are written to describe each attribute individually, and the text prompt is constructed by combining these sentences. In contrast to previous work (Guo et al., 2023), which relies on vendors to write and combine sentences, PromptTTS 2 capitalizes on the capabilities of LLM (Brown et al., 2020a; Chowdhery et al., 2022) that have demonstrated human-level performance in various tasks (Bubeck et al., 2023; Touvron et al., 2023). We instruct LLM to write high-quality sentences describing the attributes and combine the sentences into a comprehensive text prompt. This fully automated pipeline eliminates the need for human intervention in text prompt writing.

The contributions of this paper are summarized as follows:

- We design a diffusion-based variation network to model the voice variability not covered by the text prompt, alleviating the one-to-many issue in text prompt based TTS systems. During inference, voice variability can be controlled by sampling from Gaussian noise conditioned on the text prompt.
- We construct and release a text prompt dataset generated by LLM, equipped with a pipeline for text prompt generation. The pipeline produces high quality text prompts and reduces the reliance on vendors to write text prompts.

---

[2]For instance, the text prompt "Please generate a voice of a boy shouting out" can describe numerous shouting voices from boys that differ in details such as timbre.

[3]It is worth noting that reference speech is only used in training variation network but not used in inference.

- We evaluate PromptTTS 2 on a large-scale speech dataset consisting of 44K hours speech data. Experimental results demonstrate that PromptTTS 2 outperforms previous works in generating voices that correspond more accurately to the text prompt while supports controlling voice variability through sampling from Gaussian noise.

## 2 BACKGROUND

How to model voice variability has long been a crucial direction in text-to-speech (TTS) research (Wang et al., 2018; Bae et al., 2020; Bak et al., 2021). In the early stage, TTS systems primarily focus on single-speaker scenarios (Wang et al., 2017; Arık et al., 2017; Ren et al., 2019), where voice information is implicitly incorporated into neural networks. Subsequently, the need for modeling diverse voices emerges, leading to the advancement of multi-speaker TTS systems (Gibiansky et al., 2017; Chen et al., 2020; Popov et al., 2021a), in which voice variability is controlled but limited in speakers in the dataset. To adapt multi-speaker TTS systems to new speakers, few-shot adaptive TTS approaches (Chen et al., 2021; Yan et al., 2021; Huang et al., 2022) have been employed, which involve fine-tuning the multi-speaker TTS model on a limited amount of target speaker data. In contrast, zero-shot adaptive TTS models utilize in-context learning to generate new voices by exclusively modeling speaker characteristics from a speech prompt (i.e., reference speech) (Wu et al., 2022; Wang et al., 2023; Shen et al., 2023; Li et al., 2023; Le et al., 2023).

Since finding reference speech can be cumbersome and the speech data of target speaker is hard to collect or even does not exist (in the voice creation scenario), above methods on modeling voice variability is not user-friendly and scenario-limited. To achieve voice generation in a more natural and general manner, text prompt based methods have been proposed (Shimizu et al., 2023; Liu et al., 2023a), which create voices using text descriptions and require human-annotated text prompt datasets for speech. However, human-constructed datasets are often limited in scale (Guo et al., 2023) or publicly inaccessible (Yang et al., 2023) due to the associated costs. In this work, we propose a pipeline that employs LLM to generate text prompts, thereby reducing the reliance on human labor.

Given that it is impossible to comprehensively describe speech with fine-grained details (Yang et al., 2022; Qian et al., 2019; 2020) using text prompts alone, there exists the one-to-many problem in the text prompt based TTS system. Different with previous works that try to construct text prompts with more details (Guo et al., 2023; Shimizu et al., 2023), we propose the variation network to alleviate the one-to-many problem by predicting the missing information about voice variability conditioned on the text prompt with a generative (diffusion) model.

## 3 PROMPTTTS 2

In this section, we firstly give an overview on the TTS system in PromptTTS 2. Then we introduce the variation network that predicts the missing information about voice variability in the text prompt. Finally, we describe our pipeline to leverage the LLM to write the text prompt dataset.

### 3.1 OVERVIEW OF TTS SYSTEM

Figure 1a and 1b present an overview of the TTS system in PromptTTS 2. Figure 1a depicts a TTS module for synthesizing speech, with its characteristics controlled by a style module. Figure 1a skips the details for TTS module because the TTS module can be any backbone capable of synthesizing speech from phonemes. We adopt TTS backbone from Shen et al. (2023), described in Appendix C.

Figure 1b illustrates the details of the style module. During training, in line with previous works (Guo et al., 2023), we employ a BERT-based model as a text prompt encoder to extract prompt hidden. To alleviate the one-to-many mapping problem (introduced in Section 1), we utilize a reference speech encoder to model the information about voice variability not covered by the text prompt, which takes a reference speech as input and outputs a reference hidden (Shen et al., 2023; Wang et al., 2023). Since both the text prompt and reference speech can have varying lengths, we extract a fixed-length representation using cross attention (Vaswani et al., 2017) with a fixed number of query tokens for both text prompt and reference speech. More specifically, the (text) prompt representations $(P_1, ..., P_M)$ are extracted by learnable query tokens $(Q_{P_1}, ..., Q_{P_M})$, and the reference (speech)

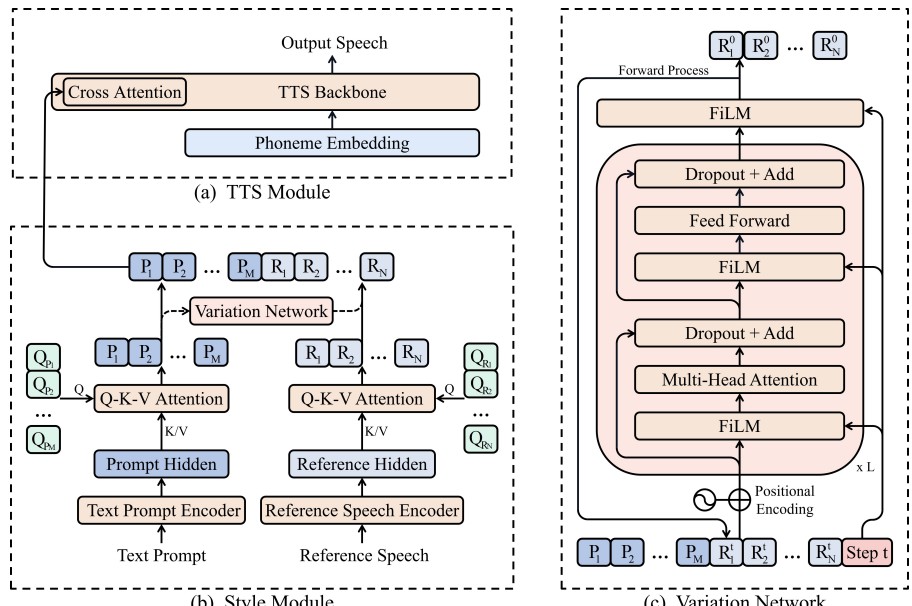

Figure 1: The overview of TTS system in PromptTTS 2. Subfigure (a) is a TTS module to synthesize speech, whose characteristics are controlled by a style module. Subfigure (b) shows the style module which takes the text prompt and reference speech as input and extracts prompt representation $(P_1, ..., P_M)$ and reference representation $(R_1, ..., R_N)$. Since the reference speech is not available in inference, we further propose a diffusion-based variation network (Subfigure (c)) to predict the reference representation based on the prompt representation.

representations $(R_1, ..., R_N)$ are extracted by learnable query tokens $(Q_{R_1}, ..., Q_{R_N})$. $M$ and $N$ represent the fixed lengths of prompt and reference representations, respectively.

During inference, only the text prompt is available, and the reference speech is not accessible, so we train a variation network to predict the reference representation $(R_1, ..., R_N)$ conditioned on the prompt representation $(P_1, ..., P_M)$, and thus the inference can be conducted with the text prompt only. The variation network is introduced in detail in the next section.

## 3.2 VARIATION NETWORK

The variation network aims to predict the reference representation $(R_1, ..., R_N)$ conditioned on the prompt representation $(P_1, ..., P_M)$. To model the reference representation, our variation network employs the diffusion model (Ho et al., 2020), which has demonstrated a robust capability in modeling multimodal distributions and complex data spaces (Kim et al., 2022; Ho et al., 2022; Nichol & Dhariwal, 2021; Leng et al., 2022). The diffusion model also enables variation network to sample different voice variability from Gaussian noise. Specifically, the diffusion model consists of a diffusion process and denoising process:

For the *diffusion process*, given the reference representation $z_0$, the forward diffusion process transforms it into Gaussian noise under the noise schedule $\beta$ as follows:

$$\mathrm{d}z_t = -\frac{1}{2}\beta_t z_t \,\mathrm{d}t + \sqrt{\beta_t}\,\mathrm{d}w_t, \quad t \in [0, 1], \tag{1}$$

For the *denoising process*, the denoising process aims to transform the noisy representation $z_t$ to the reference representation $z_0$ by the following formulation (Song et al., 2020):

$$\mathrm{d}z_t = -\frac{1}{2}(z_t + \nabla \log p_t(z_t))\beta_t \,\mathrm{d}t, \quad t \in [0, 1]. \tag{2}$$

Variation network is trained to estimate the gradients of log-density of noisy data ($\nabla \log p_t(z_t)$) by predicting the origin reference representation $z_0$ (Song et al., 2020; Shen et al., 2023), conditioned on the prompt representation, noised reference representation, and diffusion step $t$ that indicates the degree of noise in diffusion model.

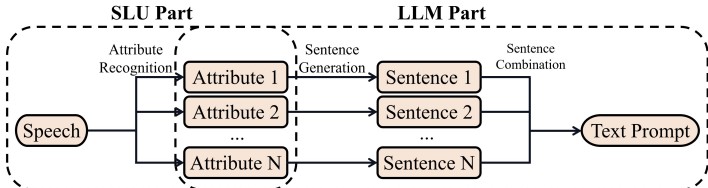

Figure 2: The overview of our prompt generation pipeline. We first recognize attributes from speech with the SLU model. Then LLM is instructed to generate sentences describing each attribute and combine the sentences of each attribute to formulate text prompts.

Figure 1c presents the detailed architecture of variation network, which is based on the Transformer Encoder (Vaswani et al., 2017). The input of variation network comprises the prompt representation $(P_1, ..., P_M)$, noised reference representation $(R_1^t, ..., P_M^t)$, and diffusion step $t$. The output of variation network is the hidden representation corresponding to the noised reference representation, optimized using L1 loss. To enhance the model's awareness of the diffusion step, we use FiLM (Perez et al., 2018) in each layer of the Transformer Encoder (Liu et al., 2023b).

In summary, during inference, we initially extract prompt representation from the text prompt using the style module. Subsequently, variation network predicts the reference representation conditioned on the prompt representation by denoising from Gaussian noise. Finally, the prompt representations are concatenated with the reference representation to guide the TTS module through cross attention.

## 3.3 TEXT PROMPT GENERATION WITH LLM

In this section, we introduce the prompt generation pipeline to build the text prompt dataset. As illustrated in Figure 2, the pipeline consists of a SLU (speech language understanding) part and a LLM (large language model) part. Given a speech, the SLU part involves tagging some labels with the speech language understanding models by recognizing attributes (e.g., gender, emotion, age) from speech; and the LLM part involves instructing large language model to write text prompts based on the labels (i.e., recognition results).

As there exist many SLU models (Baevski et al., 2020; Arora et al., 2022) to recognize attributes from speech, we focus on the LLM part for the text prompt writing based on the recognition results of SLU model. It is worth noting that text prompts written by LLM part can be reused for multiple speech with the same labels[4]. In order to improve the quality of text prompts, the LLM is instructed step by step to compose text prompts with high diversity in vocabulary and sentence format. The detail about LLM part is shown in Figure 3 and introduced as follows:

- **Keyword Construction** The SLU models recognize attributes that can describe speech characteristics. For each attribute, the SLU model recognizes several classes representing the values of the attributes. Subsequently, LLM is instructed to generate several keywords describing each class for every attribute. In the stage 1 of Figure 3, we utilize four attributes, including gender, pitch, speed, and volume. The "gender" attribute comprises two classes: male and female. The keywords generated by LLM for the male class are "man","he", and so on.

- **Sentence Construction** In addition to the variance in keywords, we also require variance in sentences. Therefore, we instruct LLM to generate multiple sentences for each attribute. A placeholder for the attribute is used by LLM when composing these sentences (e.g., word "[Gender]" is the placeholder for "gender" attribute in the stage 2 of Figure 3). The design of the placeholder offers two advantages: 1) it emphasizes the attribute for LLM, ensuring that the attribute is not omitted in the output sentence, and 2) the output sentence serves as a general template for all classes for an attribute, enabling the generation of diverse text prompts by filling the placeholder with different keywords. In the provided example, the stage 2 of Figure 3 illustrates several sentences composed by LLM that describe different attributes.

- **Sentence Combination** Since text prompts can describe more than one attribute, we perform sentence combination based on the sentences generated in the stage 2. LLM is instructed to combine sentences describing different attributes into a new sentence, allowing us to obtain text

---

[4]Since the recognition results of SLU models are in a pre-defined label set.

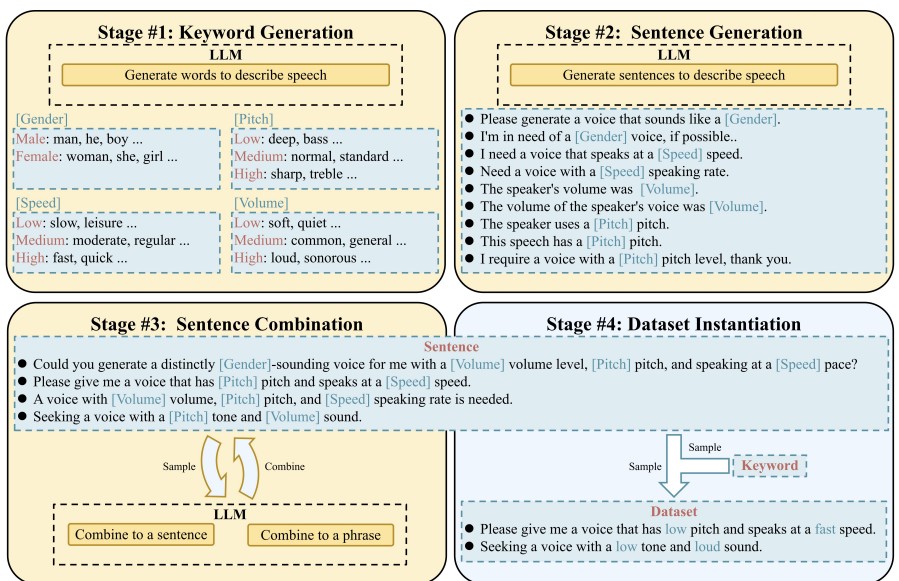

Figure 3: Text prompt generation using LLM: In Stage 1, LLM generates keywords for each attribute (gender, pitch, speed, and volume). In Stage 2, LLM composes sentences for each attribute, integrating placeholders for the corresponding attributes. In Stage 3, LLM combines the sentences from Stage 2 to create a sentence that simultaneously describes multiple attributes. In Stage 4, the dataset is instantiated by initially sampling a combined sentence and subsequently sampling keywords to replace the placeholders within the sentence.

prompts representing various combinations of attributes. It is worth noting that the sentences generated by LLM are always complete and free of grammatical errors. In contrast, users of text prompt based TTS systems may not always describe voices in a formal manner. Consequently, we also instruct LLM to write phrases to enhance the diversity of constructed sentences. In the stage 3 of Figure 3, we present some example combination sentences and phrases generated by LLM.

- **Dataset Instantiation** The results generated from the previously described three stages form the final text prompt dataset, which is employed alongside a speech dataset. For each instance of speech $S$ within the speech dataset, we tag a class label on every attribute with SLU models. Following this, we select a sentence that encompasses all the attributes of speech $S$. Next, we obtain a keyword for each attribute of speech $S$ based on its corresponding class label. The ultimate text prompt is instantiated by substituting all placeholders in the sentence with their corresponding keywords. In the stage 4 of Figure 3, we provide examples illustrating the finalized text prompts. The speech $S$ and the corresponding finalized text prompt formulate a speech-prompt paired data.

We provide an example of our pipeline in Appendix A, which shows the dialogue process with LLM. More discussion about the scalability of our pipeline can be found in Appendix B.

## 4 EXPERIMENT CONFIGURATION

In this section, we present the experimental configurations, including the datasets, TTS backbone, baseline systems and experiment details.

**Datasets** For the speech dataset, we employ the English subset of the Multilingual LibriSpeech (MLS) dataset (Pratap et al., 2020), which comprises 44K hours of transcribed speech data from LibriVox audiobooks. For the text prompt data, we utilize PromptSpeech (Guo et al., 2023) that contains 20K text prompts written by human describing speech from four attributes including pitch, gender, volume, and speed. We also utilize our prompt generation pipeline to write 20K text prompts with the help of LLM (GPT-3.5-TURBO). The test set of PromptSpeech is used as test data, which contains 1305 text prompts. For the SLU model on attribute recognition, we identify gender using an

Table 1: The accuracy (%) of synthesized speech on the attribute control of PromptTTS 2 and baselines.

| Model | Gender | Speed | Volume | Pitch | Mean |
|---|---|---|---|---|---|
| PromptTTS (Guo et al., 2023) | 98.01 | 89.66 | 92.49 | 85.98 | 91.54 |
| InstructTTS (Yang et al., 2023) | 97.24 | 90.57 | 91.26 | 86.82 | 91.47 |
| PromptTTS 2 | **98.23** | **92.64** | **92.56** | **89.89** | **93.33** |

open-source model[5], and the other attributes (i.e., pitch, volume, and speed) are recognized using digital signal processing tools[6].

**TTS Backbone**    In general, PromptTTS 2 extracts a fixed-dimension representation to control the characteristics of synthesized speech. This approach can be incorporated into any TTS backbone by integrating the representations into the TTS backbone with cross attention. We adopt TTS backbone from a SOTA TTS system, NaturalSpeech 2 (Shen et al., 2023), whose details are in Appendix C.

**Baseline Systems**    We compare PromptTTS 2 with current SOTA systems of text prompt based TTS, PromptTTS (Guo et al., 2023) and InstructTTS (Yang et al., 2023). To ensure a fair comparison, we modify the backbone in baseline systems to the latent diffusion backbone used in PromptTTS 2.

**Experiment Details**    The number of layers in the reference speech encoder and variation network is 6 and 12, respectively, with a hidden size of 512. The query number $M, N$ in style module is both set to 8. Concerning the TTS backbone and the text prompt encoder, we adhere to the settings in NaturalSpeech 2 (Shen et al., 2023) and PromptTTS (Guo et al., 2023), respectively. The training configuration is also derived from NaturalSpeech 2 (Shen et al., 2023).

## 5    RESULT

In this section, we evaluate the effectiveness of PromptTTS 2. Firstly, We compare the accuracy of attribute control and the speech quality between PromptTTS 2 and baseline systems in Section 5.1. In Section 5.2, we demonstrate that the variation network successfully captures the information about voice variability. In Section 5.3, we compare the text prompts generated by our pipeline with those written by human or other LLM based method. Finally, we conduct an analysis on the style module in Section 5.4 and perform an extension on face-to-voice (Face2Voice) generation in Section 5.5.

### 5.1    EFFECTIVENESS OF PROMPTTTS 2

We evaluate the effectiveness of PromptTTS 2 from the perspective of attribute control and speech quality. First, we compare the accuracy of attribute control between PromptTTS 2 and baseline systems, presented in Table 1. The results demonstrate that PromptTTS 2 can synthesize speech with higher accuracy across all attributes compared to baseline systems, achieving an average improvement of 1.79%. In Table 1, the experiments use the text prompts from our pipeline, and more results on different text prompts can be found in Appendix D. Then we conduct mean-of-score (MOS) and comparative MOS (CMOS) test to evaluate the speech quality of PromptTTS 2 and baseline systems, as shown in Table 2. The results of MOS and CMOS show that PromptTTS 2 achieves higher speech quality than the baseline systems.

### 5.2    STUDY OF VARIATION NETWORK

We examine the information of voice variability learned by variation network. Due to the one-to-many problem between the text prompt and the voice variability in speech, the model might implicitly incorporate voice variability information into specific *aspects*. Consequently, the model could synthesize varying voices even when presented with identical text prompts (or text prompts with equivalent meanings). For the baseline systems, PromptTTS and InstructTTS, these *aspects* include

---

[5]https://github.com/karthikbhamidipati/multi-task-speech-classification
[6]https://github.com/JeremyCCHsu/Python-Wrapper-for-World-Vocoder

Table 2: The results of speech quality with 95% confidence intervals. GT stands for the recording. Codec reconstruction stands for that the waveform is encoded to latent representation first and then reversed to waveform by the decoder of codec.

| Setting | MOS | CMOS (vs. PromptTTS 2) |
|---|---|---|
| GT | $4.38 \pm 0.08$ | - |
| GT (Codec Reconstruction) | $4.30 \pm 0.07$ | - |
| PromptTTS (Guo et al., 2023) | $3.77 \pm 0.09$ | -0.191 |
| InstructTTS (Yang et al., 2023) | $3.80 \pm 0.07$ | -0.157 |
| PromptTTS 2 | $\mathbf{3.88 \pm 0.08}$ | **0.0** |

Table 3: The average speech similarity of systems when synthesizing speech with the same intention in text prompts but different text prompts, text contents, sampling results of TTS backbone and sampling results of variation network. The similarity score is in a range of [0, 1].

| Model | Text Prompt | Text Content | TTS Backbone | Variation Network |
|---|---|---|---|---|
| PromptTTS | 0.766 | 0.662 | 0.799 | - |
| InstructTTS | 0.773 | 0.718 | 0.796 | - |
| PromptTTS 2 | 0.775 | 0.873 | 0.914 | 0.355 |

the text prompt (with the same meaning), text content, and TTS backbone (with latent diffusion), as the voice of synthesized speech may differ depending on the text prompt, text content, and TTS backbone. In PromptTTS 2, an additional *aspect*, variation network, is introduced, as the voice of synthesized speech may also vary based on different sampling results of the variation network.

We use WavLM-TDNN model (Chen et al., 2022a) to assess the similarity of two speech in a range of [0, 1], where the higher speech similarity, the less voice variability. For each *aspect* mentioned above, we generate 5 speech and calculate the average similarity of the 5 speech. The results are shown in Table 3. From the table, we have the following observation: 1) baseline systems implicitly acquire a small amount of voice variability information in the aspect of the text prompt, text content, and TTS backbone, which is undesired as we aim for style to be controlled exclusively by the intention in text prompt; 2) the speech similarity of variation network in PromptTTS 2 is markedly lower than other aspects, showing that the variation network effectively models voice variability information not encompassed by the text prompts (i.e., different sampling results leads to different timbre); 3) for PromptTTS 2, the voice variability acquired in aspects apart from variation network is less than those of baseline systems whose similarity are higher. This indicates that when the variation network successfully captures voice variability, the model is inclined to learn less voice variability information in other aspects. We strongly encourage readers to listen to the samples on our demo page, which offer an intuitive comprehension of the voice variability information present in each dimension.

Besides the WavLM-TDNN model, we evaluate the speech similarity by human experts. The conclusions of subjective test are similar with those of WavLM-TDNN model, shown in Appendix E.

## 5.3 PROMPT GENERATION QUALITY

We analyze the quality of text prompts generated by our pipeline through whether the text prompts can reflect the values of attributes. Specifically, we train a classifier to recognize the intention of text prompts on four attributes. The training data for the classifier is 1) text prompts authored by human (i.e., the training set of PromptSpeech (Guo et al., 2023)), 2) TextrolSpeech (Ji et al., 2023) whose text prompts are written by LLM (GPT-3.5-TURBO) with multi-stage prompt programming approach (but without the placeholder or sentence combination mechanism in our pipeline), 3) text prompts written by our pipeline. We display the average accuracy of classification on the test set of PromptSpeech in Table 4. The classifier trained on text prompts generated by our pipeline has a higher accuracy compared to the classifier trained on text prompts authored by human or TextrolSpeech. This result indicates that the text prompts generated by our pipeline exhibit higher quality than previous works, verifying the effectiveness of our prompt generation pipeline. More ablation studies on our prompt generation pipeline can be found in Appendix F.

Table 4: The accuracy (%) of intention classification on four attributes with text prompts from PromptSpeech, TextrolSpeech, and our prompt generation pipeline.

| Training Set | Gender | Speed | Volume | Pitch | Mean |
|---|---|---|---|---|---|
| PromptSpeech (Guo et al., 2023) | **100.00** | 96.85 | 89.58 | 84.51 | 92.74 |
| TextrolSpeech (Ji et al., 2023) | 98.77 | 94.18 | 93.10 | 92.80 | 94.71 |
| Our Prompt Generation Pipeline | 99.08 | **97.47** | **94.48** | **94.48** | **96.38** |

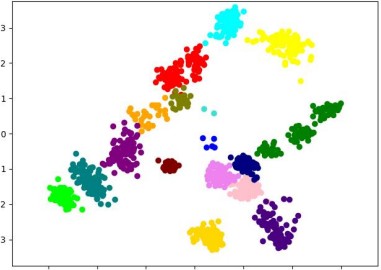 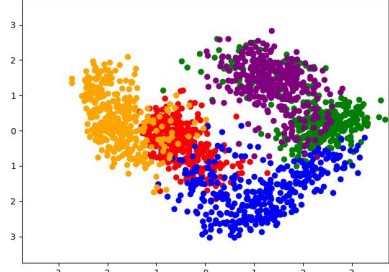

Figure 4: The PCA results of the representation extracted by the reference speech encoder in style module. Each point stands for a speech and the speech with the same *speaker* (left figure) or the same same *emotion* (right figure) has the same color.

## 5.4 FURTHER ANALYSIS

In this section, we study the reference representation extracted from reference speech encoder in style module, which is a high-dimensional vector. To visualize the vector, we employed Principal Component Analysis (PCA) to reduce the dimensionality of the vector and map it to a 2D vector, which is plotted in Figure 4. Each point in figure stands for a speech and the speech with the same *speaker* or the same *emotion* (Zhou et al., 2021; 2022) has the same color. We observe that the speech samples belonging to the same speaker or the same emotion tend to cluster together in the figure. This observation suggests that the reference representations effectively learn the voice variability uncovered by text prompts (such as speaker or emotion). Therefore, given a text prompt, the variation network can sample different voice variability corresponding to the text prompt, which offers users more flexibility on generating voices. More ablations on PromptTTS 2 are in Appendix G and H.

## 5.5 EXTENSION ON FACE2VOICE

PromptTTS 2 involves modeling voice information utilizing a sequence of predictable tokens, enabling its extension to many other scenarios involving predicting voices from other modalities. We conduct a preliminary experiment on the Face2Voice extension, with a objective of predicting voices based on speaker's facial images. More details about Face2Voice extension can be found in Appendix I, which shows that PromptTTS 2 generates voices corresponding more closely to the facial images compared with the baseline method (Weng et al., 2023). Furthermore, our findings show that PromptTTS 2 is a general method for generating voices conditioned on text prompts, facial images, or other information. Samples of facial images and generated voices can also be found on our demo page.

## 6 CONCLUSION

In this work, we propose PromptTTS 2 to address the one-to-many and data-scale issues in text prompt based TTS systems, which implements a variation network to model the voice variability information not captured by text prompts and uses LLM for high-quality text prompt generation. The variation network facilitates more detailed voice control by sampling from Gaussian noise. The LLM-based prompt generation pipeline eliminates the reliance on vendors and provides scalability for easily incorporating new attributes. Experimental results indicate that the proposed variation network assists the TTS model in synthesizing speech more closely corresponding to the text prompt and diverse in voice variability. Our pipeline generates text prompts with higher quality than human-authored ones. For future work, we plan to extract additional attributes from large-scale speech data to increase the diversity of voice generation system and apply our method on more modalities for voice generation.

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

## A EXAMPLE OF PROMPT GENERATION PIPELINE

Table 5 shows a detailed example of our prompt generation pipeline including the queries to instruct LLM. In Stage 1, we ask LLM to generate keywords for "male gender". In Stage 2, we ask LLM to write sentences describing "pitch" attribute, using "[placeholder]" to stand for the corresponding attribute (i.e., "pitch"). In Stage 3, we ask LLM to combine two sentences describing "gender" attribute and "pitch" attribute into one sentence. In Stage 4, we construct a text prompt by first sampling a sentence and subsequently sampling keywords to replace the placeholders within the sentence.

Table 5: The example of a text prompt generated by LLM with the pipeline.

| Stage | Dialogue |
|---|---|
| Stage 1 | Query: Can you list 5 words or phrases that is the synonyms for male gender in lower case? Please generate one word per line and identify it with a serial number.
Response: 1. man.
2. boy.
3. guy.
4. dude.
5. gentleman. |
| Stage 2 | Query: Please generate 5 templates to to ask for generating a voice. These templates can only describe the pitch of the voice and use [placeholder] to indicate where a word to describe pitch would be inserted. This template should be simple enough to only have a few words.
Response: 1. I need a [placeholder] voice.
2. Looking for a [placeholder] voice.
3. In search of a [placeholder] voice.
4. Seeking a voice with [placeholder] pitch.
5. Require a voice, [placeholder] pitch. |
| Stage 3 | Query: I have some sentences, can you combine these sentences into one phrase and keep all of the '[]' to describe the style of speech with the same meaning? You can generate 5 different phrases and callout number. Please reduce to phrases. The sentences are: Please generate a voice that sounds like a [Gender]. Seeking a voice with [Pitch] pitch.
Response: 1. Generate a voice that sounds like a [Gender] with [Pitch] pitch.
2. Looking for a voice that embodies a [Gender] with a [Pitch] pitch.
3. Require a voice with a [Pitch] pitch that represents a [Gender].
4. Requesting a [Gender] voice at [Pitch] pitch.
5. Desiring a [Gender] voice with a [Pitch] pitch. |
| Stage 4 | Generate a voice that sounds like a boy with low pitch. |

## B SCALABILITY OF PROMPT GENERATION PIPELINE

We present a brief discussion on the scalability of our prompt generation pipeline. With the help of our pipeline, incorporating a new attribute requires only the definition of classes for the new attribute and the tagging of the speech dataset for that attribute using a SLU model (Baevski et al., 2020; Arora et al., 2022). For example, if we intend to introduce a new "age" attribute into the pipeline, we can define three classes corresponding to the "age" attribute, namely "teenager", "adult" and "elder". Subsequently, the pipeline can generate a text prompt dataset for the "age" attribute with the help of LLM and a SLU model on "age" attribute to tag the speech dataset. In summary, our pipeline simplifies the process of adding new attributes, allowing for easier expansion and adaptability to diverse speech characteristics.

## C DETAILS ON THE TTS BACKBONE

The TTS backbone of PromptTTS 2 is adopted from a state-of-the-art large-scale TTS system, NaturalSpeech 2 (Shen et al., 2023), which consists of 1) a neural audio codec that transforms the audio waveform into latent vectors and reconstructs the latent representation into the waveform, and

Table 6: The accuracy (%) of synthesized speech on the attribute control of PromptTTS 2 and baselines when using the text prompt from PromptSpeech (Guo et al., 2023).

| Model | Gender | Speed | Volume | Pitch | Mean |
|---|---|---|---|---|---|
| PromptTTS (Guo et al., 2023) | **98.93** | 87.43 | 89.35 | 85.44 | 90.29 |
| InstructTTS (Yang et al., 2023) | 96.55 | 86.13 | 88.43 | 85.52 | 89.16 |
| PromptTTS 2 | 98.77 | **90.80** | **90.57** | **89.58** | **92.43** |

Table 7: In human subjective test, the average speech similarity (%) of baseline systems and PromptTTS 2 when synthesizing speech with the same intention in text prompts but different text prompts, text contents, sampling results of TTS backbone and sampling results of variation network.

| Model | Text Prompt | Text Content | TTS Backbone | Variation Network |
|---|---|---|---|---|
| PromptTTS | 94.44 | 79.63 | 96.30 | - |
| InstructTTS | 92.59 | 85.18 | 94.44 | - |
| PromptTTS 2 | 90.74 | 98.00 | 98.15 | 7.41 |

2) a latent diffusion model with a prior (a duration/pitch predictor and a phoneme encoder). In detail, we first encode the audio waveform into a latent representation using the residual vector-quantizer (RVQ) (Zeghidour et al., 2021). Then, the latent diffusion denoises (synthesizes) the latent speech representation from Gaussian noise. The denoised latent representation is subsequently converted back to the waveform by the decoder of the neural audio codec.

## D    ABLATION ON TTS SYSTEMS WITH DIFFERENT TEXT PROMPT

Since the ablation study on the text prompt shows the superiority of the text prompts from our prompt generation pipeline over those in other baseline methods (as reported in Table 4), the results in Table 1 are conducted when all the models using the text data from our prompt generation pipeline. Thus, the results are a fair comparison in terms of text prompts.

To strengthen the conclusions of our paper, we conducted additional experiments on all models (i.e., PromptTTS, InstructTTS, and PromptTTS 2) using the text data in PromptSpeech (Guo et al., 2023). The results are shown in Table 6.

From the table, we observe that PromptTTS 2 outperforms baseline methods in average on Prompt-Speech text prompt datasets. By further taking the results in Table 1 into consideration, we find that using text data from our pipelines improves the quality of all text prompt based TTS models on most of attributes, compared to using the prompts in PromptSpeech.

## E    SUBJECTIVE TEST ON THE VOICE VARIABILITY IN VARIATION NETWORK

Besides the metric by WavLM-TDNN model, we also evaluate the speech similarity from the perspective of human. For each *aspect* mentioned in Section 5.2, we generate 5 speech and calculate the average similarity of the 5 speech. In human subjective test, the judges are asked to judge whether the two synthesized speech are in the same style. The speech similarity of each *aspect* is defined as the ratio of speech pair (among the 5 speech) that is regarded as in the same style by judges. The conclusions of subjective test (Table 7) are similar with those of WavLM-TDNN model discussed in Section 5.2.

## F    ABLATION STUDY ON PROMPT GENERATION PIPELINE

We conduct ablation studies on the prompt generation pipeline. First, we remove the design of the placeholder from the pipeline. In this case, LLM is required to directly write text prompts for each class in attributes, after which sentence combination is performed. The results are presented as "- Placeholder" in Table 8. The drop in classification accuracy demonstrates that the placeholder is beneficial for the prompt generation pipeline. Without it, LLM might miss attributes or even

Table 8: The accuracy (%) of intention classification on four attributes in the ablation of our prompt generation pipeline.

| Datasets | Gender | Speed | Volume | Pitch | Mean |
|---|---|---|---|---|---|
| Our Prompt Generation Pipeline | 99.08 | 97.47 | 94.48 | 94.48 | 96.38 |
| - Placeholder | 99.08 | 97.31 | 89.27 | 90.50 | 94.04 |
| - Phrase | 99.08 | 97.01 | 95.55 | 92.72 | 96.09 |
| - Sentence | 99.08 | 97.47 | 93.18 | 94.94 | 96.16 |

Table 9: The accuracy (%) of synthesized speech on the attribute control of PromptTTS 2 and baselines when using the TTS backbone from SoundStorm (Borsos et al., 2023).

| Model | Gender | Speed | Volume | Pitch | Mean |
|---|---|---|---|---|---|
| PromptTTS (Guo et al., 2023) | 97.47 | 89.96 | 92.41 | 85.06 | 91.23 |
| InstructTTS (Yang et al., 2023) | 96.55 | 89.66 | 92.18 | 84.14 | 90.63 |
| PromptTTS 2 | **98.92** | **92.18** | **93.93** | **90.73** | **93.95** |

alter them during sentence combination, resulting in low-quality text prompts. In addition to the placeholder, we also conduct ablation studies on instructing LLM to write only phrases or sentences by removing sentences ("- Sentence") or phrases ("- Phrase"). The results indicate that variations in format can marginally improve the robustness of the prompt generation pipeline.

## G    ABLATION ON TTS BACKBONE

Besides the TTS backbone based on latent diffusion (Shen et al., 2023), we further apply PromptTTS 2 (as well as the baseline methods) on another TTS backbone based on the token prediction of codec results in SoundStorm (Borsos et al., 2023). The results are in Table 9:

From the result, we observe that PromptTTS 2 consistently outperform baseline methods in the condition that TTS backbone in SoundStorm (Borsos et al., 2023) is leveraged.

## H    ABLATION ON REPRESENTATION LENGTH

We further conduct ablation study on a unique hyper-parameter in PromptTTS 2, i.e., the length of prompt and reference representations. The results are shown in Table 10. The results indicate that representation length is not a highly sensitive hyper-parameter in terms of performance, and increasing the length to 16 can lead to a slight improvement in model accuracy.

## I    EXTENSION ON FACE2VOICE

PromptTTS 2 involves modeling voice information utilizing a sequence of predictable tokens, enabling its extension to many other scenarios involving predicting voice from other modalities.

We conduct a preliminary experiment on the Face2Voice extension, with a objective of predicting voice based on the facial image of speaker. In this experiment, the facial image is processed using an

Table 10: The accuracy (%) of synthesized speech on the attribute control of PromptTTS 2 when using different lengths of the prompt and reference representations.

| Representation Length | Gender | Speed | Volume | Pitch | Mean |
|---|---|---|---|---|---|
| 4 | 98.54 | 92.49 | 93.26 | 88.66 | 93.24 |
| 8 (Default Choice) | 98.23 | 92.64 | 92.56 | 89.89 | 93.33 |
| 16 | 98.77 | 92.18 | 93.95 | 90.27 | 93.79 |
| 32 | 98.39 | 91.80 | 91.19 | 90.27 | 93.09 |

Table 11: The MOS results (%) on whether the voice is in the same style with the facial image or not. GT stands for judging whether the ground-truth voice is in the same style with the corresponding facial image.

| Setting | Same | In-between | Different |
|---|---|---|---|
| GT | 46.47 | 42.05 | 11.47 |
| SP-FaceVC (Weng et al., 2023) | 20.17 | 45.38 | 34.45 |
| PromptTTS 2 | 31.78 | 41.17 | 27.05 |

Table 12: The CMOS results (%) on which voice (synthesized by PromptTTS 2 or SP-FaceVC) corresponds more closely with the facial image.

| Setting | Former | Tie | Latter |
|---|---|---|---|
| PromptTTS 2 vs. SP-FaceVC (Weng et al., 2023) | 51.47 | 29.41 | 19.12 |

image encoder[7] pretrained in CLIP (Schuhmann et al., 2022; Radford et al., 2021; Ilharco et al., 2021) to extract image representations. Simultaneously, the speech is processed using a reference speech encoder depicted in Figure 1b to extract reference representations. Subsequently, a variation network (illustrated in Figure 1c) is trained to predict reference representations from image representations.

For this preliminary experiment, we utilize the HDTF dataset (Zhang et al., 2021), a high-resolution dataset designed for talking face generation. The dataset includes more than 300 distinct speakers and encompasses 15.8 hours of video. To extract paired data of facial images and speech, we first select an image (video frame) and then extract a speech segment with a duration of 5-10 seconds surrounding the chosen frame. We designate 18 speakers for testing and use the remaining speakers for training.

We compare our method with a SOTA method on Face2Voice, SP-FaceVC (Weng et al., 2023)[8], with subjective test (MOS). In the MOS test, the judges are asked to judge whether a facial image and the voice is in the same style (i.e., it is natural for the facial image to have that voice), whose results are shown in Figure 11. The results demonstrate that compared with SP-FaceVC, PromptTTS 2 can generate voice corresponding more closely with the facial image (31.78% versus 20.17%) and fewer unsuitable cases (27.05% versus 34.45%).

We also conduct comparative MOS (CMOS) test to directly judge that given a facial image, which voice (synthesized by PromptTTS 2 or SP-FaceVC) corresponds more closely with the facial image. The results in Table 12 show that in 80.88% cases, PromptTTS 2 synthesizes a better or comparable voice than SP-FaceVC. Furthermore, our findings demonstrate that PromptTTS 2 is a general method for generating voices conditioned on text prompts, facial images, or other types of information. Samples of facial images and generated voices can also be found on our demo page.

---

[7]https://github.com/mlfoundations/open_clip
[8]https://github.com/anitaweng/SP-FaceVC

