# OpenReview forum: "PromptTTS 2: Describing and Generating Voices with Text Prompt"
_ICLR.cc/2024/Conference — ICLR 2024 poster_

### Official Review · Reviewer_dNEb · 2023-10-28

**Soundness:** 3 good
**Presentation:** 3 good
**Contribution:** 2 fair
**Rating:** 6
**Confidence:** 4

**Summary:**

The paper proposes a small architecture improvement upon PromptTTS and also proposes to use SLU and LLM to generate text prompts.

**Strengths:**

1. There is some novelty in using SLU and LLM to generate text prompts for voice, as I believe this is the first work investigating this.
2. A good mount of ablation and analysis.
3. The paper is in general written clearly and easy to follow.

**Weaknesses:**

1. The authors claim that the proposed variational method solves the one-to-many problem, but I doubt if this is the case. The proposed module basically learns to predict a melspec encoder's output from text prompts, in addition to conditoning the TTS backbone. I don't see how this solves the one-to-many problem as in the end we are still predicting from text (one) to different variability (many) in the voice. The only difference is that there is an additional auxiliary loss from the melspec encoder. This seems to be in principle similar to InstructTTS's approach to regularize melspec encoder and text encoder output to close in the embedding space.
2. The proposed prompt generation method, while the first of its kind, seems to be still quite limited by the very limited SLU output set, and the template used for LLM generation.
3. The proposed method performs only marginally better than PromptTTS and InstructTTS.

**Questions:**

Will the author open source the generated dataset or source code? I would consider the open sourcing effort as a contribution.

---

> ### Author Response · Authors · 2023-11-19
> **Response to Reviewer dNEb**
>
> We sincerely appreciate your efforts in reviewing our paper and providing us with valuable and constructive feedback. Your comments have greatly benefited our work. We have addressed your questions below.
>
> **Q1: About one-to-many problem**
>
> As mentioned in your review, previous work such as InstructTTS has attempted to address the one-to-many problem by regularizing the melspec encoder and text encoder outputs to converge in the embedding space using L2 loss. However, this approach is limited in its ability to resolve the one-to-many problem, as the output of the melspec encoder contains much more information than that of the text encoder.
>
> In contrast, we approach the one-to-many problem as a generation task and utilize generative models (specifically, diffusion models) to generate the output of the melspec encoder based on the output of the text encoder. We believe that VoiceGen differs from previous frameworks, and is the first framework capable of generating and fixing virtual speakers. This is a significant contribution to both future research and the real-world application of prompt-based TTS systems.
>
> **Q2: About the limitation of SLU output set and template**
>
> The template is written by LLM and can be various by changing the prompt. However, the current limitations of the SLU output set can restrict the capabilities of VoiceGen. We anticipate that more robust SLU models will become available in the future, and we plan to continue our efforts to improve the performance of VoiceGen in this regard.
>
>
> **Q3: About the performance**
>
> Firstly, our results demonstrate that VoiceGen outperforms previous methods such as PromptTTS and VoiceGen by a significant margin. Further analyses (see Q1 in [Response to Reviewer g23M](https://openreview.net/forum?id=NsCXDyv2Bn&noteId=j7nT1dF7U1) and Q7 in [Response to Reviewer Ghqv (2/2)](https://openreview.net/forum?id=NsCXDyv2Bn&noteId=w6K4vz3QGP)) have confirmed that this improvement is consistent across different datasets and TTS backbones.
>
> Secondly, the main contribution of our paper lies in the development of a general framework on modeling voice variability (first work to support virtual speaker creation) and text prompt writting (to save labeling cost while achieving good quality). We sincerely hope that the significance of our contribution is recognized.
>
> **Q4: About the code and dataset**
>
> We have updated the supplementary material to include the code for VoiceGen and the prompt generation pipeline, along with the generated dataset. It is worth noting that this pipeline can be utilized to generate prompts for other attributes by providing the necessary attributes, classes, and ChatGPT keys (further details can be found in the README file located in the data_pipeline folder).
>
> **Finally, we would like to express our gratitude again to the reviewer for their time and effort in reviewing our paper. Please do not hesitate to let us know if you have any further concerns or comments. We would be happy to address them.**

---

> > ### Author Response · Authors · 2023-11-22
> > **Looking forward to further discussion**
> >
> > Dear Reviewer dNEb,
> >
> > As the deadline for discussion is approaching, we would like to inquire that whether we have effectively addressed the questions raised in the initial review. If you have any further question, we would be very happy to reply.
> >
> > Thank you again for your time and effort in reviewing.
> >
> > Best,
> > Authors

---

> > > ### Comment · Reviewer_dNEb · 2023-11-22
> > >
> > > Thanks for the reply. I am still not convinced by the reply to Q1. I don't think it solves the one-to-many problem. I can only see that adding an intermediate representation which the text encoder learns to predict helps with the performance. However since the authors addressed other issues and are open-sourcing their efforts. I would like to raise my score slightly.

---

> > > > ### Author Response · Authors · 2023-11-23
> > > > **Thanks for your comments**
> > > >
> > > > Dear Reviewer dNEb,
> > > >
> > > > We are very delightful to see that most of the issues are addressed. For the Q1, we agree that "solve" the one-to-many problem is a kind of over-claim, as VoiceGen propose a better framework to "alleviate" the one-to-many problem (by using a generative model to predict the missing information about voice variability) than the previous works (using L2 regularization). We have changed the statements in our paper from "solve" the problem to "alleviate" the problem. Thanks again for your comments.
> > > >
> > > > Best, Authors

---

### Official Review · Reviewer_2wDM · 2023-10-30

**Soundness:** 3 good
**Presentation:** 3 good
**Contribution:** 3 good
**Rating:** 6
**Confidence:** 5

**Summary:**

In this paper, the authors proposed VoiceGen, a text-to-speech framework that uses a variation network to provide variability information of voice not captured by text prompts, and a prompt generation pipeline to utilize the large language models (LLM) to compose high quality text prompts.

The variation network predicts the representation extracted from the reference speech based on the text prompt. And the LLM formulates text prompts based on the recognition results from speech language understanding model. Compared to the previous works, VoiceGen generates voices more consistent with text prompts, offering users more choices on voice generation.

**Strengths:**

The proposed modeling and data labeling pipelines for text-prompt based TTS systems can generate higher-quality speech with more consistent and noticeable control compared to previous systems. The variation network predicts speech representations that are more closely corresponding to the text prompt and more diversity by sampling from Gaussian noise. On the other hand, the LLM-based prompt generation pipeline can produce high-quality text prompts at scale and can easily incorporate new attributes. Overall, the proposed system provides a framework that is beneficial for future text-prompt based TTS research.

**Weaknesses:**

After listening to the generated voices on the demo page, audio quality is still an issue and further improvements are required, especially for certain text prompts such as "Please speak at a fast speed, gentleman". The reason could be missing or few audio samples for corresponding prompts in training datasets.

**Questions:**

1. What's the necessity of concatenating both text prompt representation and speech prompt representation for TTS backbone? Will the speech prompt representation itself be enough to guide the TTS backbone through cross attention?

2. What's the impact of making use of a fixed-length representation for text and speech prompt representations? Fixed-length representations may be fine for global attributes such as age and gender, but is that enough for fine-grained or local control in guided speech generation?

---

> ### Author Response · Authors · 2023-11-19
> **Response to Reviewer 2wDM**
>
> We sincerely appreciate your efforts in reviewing our paper and providing us with valuable and constructive feedback. Your comments have greatly benefited our work. We have addressed your questions below.
>
> **Q1: About imperfect sample**
>
> Yes, we keep this imperfect sample to showcase that the sampling result can be occasionally imperfect (although this can be alleviated by sampling for another time). We will keep working on further improving the stability and quality of model in future work.
>
> **Q2: About the concatenating of both text prompt representation and speech prompt representation**
>
> As you noted in your review, using only speech prompt representation is sufficient in terms of information completeness. However, concatenating both text prompt representation and speech prompt representation can enhance the model's awareness of the information contained within the text prompt, thereby leading to a slight improvement in model accuracy.
>
> **Q3: About the fixed-length representation for text and speech prompt representations**
>
> We conduct experiments on the this hyper-parameter in VoiceGen, the length of prompt and reference representations. The results are shown as follows:
>
> |  Representation Length   | Gender | Speed | Volume | Pitch | Mean |
> |  ----  | ----  |  ----  | ----  |  ----  | ----  |
> | 4  | 98.54 | 92.49 | 93.26 | 88.66 | 93.24 |
> | 8 (Default Choice)  | 98.23 | 92.64 | 92.56 | 89.89 | 93.33 |
> | 16  | 98.77 | 92.18 | 93.95 | 90.27 | 93.79 |
> | 32  | 98.39 | 91.80 | 91.19 | 90.27 | 93.09 |
>
> The results indicates that representation length is not a highly sensitive hyper-parameter in terms of performance, and increasing the length to 16 can lead to a slight improvement in model accuracy.
>
> We believe that variable-length representation could enhance the model's fine-grained or local control, particularly as the text prompt becomes more complex. We plan to explore this area further in future research.
>
> **Finally, we would like to express our gratitude again to the reviewer for their time and effort in reviewing our paper. Please do not hesitate to let us know if you have any further concerns or comments. We would be happy to address them.**

---

### Official Review · Reviewer_Ghqv · 2023-10-30

**Soundness:** 2 fair
**Presentation:** 2 fair
**Contribution:** 2 fair
**Rating:** 6
**Confidence:** 3

**Summary:**

The paper is about text-to-speech (TTS). The TTS model is based on Naturalspeech 2. The TTS model is extended by a style module, which uses a text prompt to describe the style. During training, it additionally also uses the reference speech as input. However, for speech generation, a variation network instead generates the reference speech encoder outputs. This variation network is the core novelty proposed by the authors. It is supposed to add further variation in the speech style which cannot be covered by the text prompt alone. The variation network is a diffusion model using a Transformer encoder to iterate on the speech encoder outputs.

This proposed model is called VoiceGen.

The training data is based on the Multilingual LibriSpeech (MLS) dataset with 44K hours of transcribed speech. To generate the text prompt, needed to train the style model, a text prompt dataset generation pipeline is proposed, to extend the given transcribed speech: Based on a speech language understanding (SLU) model, the gender of the speech is identified. Additionally, using digital signal processing tools, pitch, volume, and speed is extracted and put into classes. Those attribute classes are then fed into a large language model (LLM) to generate a text prompt which conveys the style attributes.

The variability is measured using a WavLM-TDNN model to assess the similarity of two speeches, and it is shown that the introduction of the variation network leads to higher speech variability. At the same time, mean-opinion-score (MOS) on the quality of the proposed VoiceGen model is slightly better than other text-prompt-based TTS models, PromptTTS and InstructTTS specifically. It is also shown that the text prompt indeed works and can generate speech with the requested attributes.

**Strengths:**

The claims are tested and it seems the proposed model adds quite a bit of variability, as it was intended.

**Weaknesses:**

So many details are left out, as this would not really be possible to fit into the paper. So without the exact code + recipe to produce all the results, it will be almost impossible to reproduce the results. I think having code + recipe available here is very important.

All the experiments basically just show that the proposed model works well and solves the outlined problem. However, there is almost no analysis or ablation studies, etc. E.g. how important is it to use a diffusion model here? What about other model types? What about other smaller model details, and hyper parameters, etc?

There are a few things a bit unclear (see below).

**Questions:**

> Compared to traditional text-to- speech (TTS) methods relying on speech prompts (reference speech) for voice variability, using text prompts (descriptions) is more user-friendly since speech prompts can be hard to find or may not exist at all.

I don't exactly understand the difference between speech prompts and text prompts.

A text prompt is really like a description of what should be generated, like “Please generate a voice of a boy shouting out”.

A speech prompt is the same but as speech? Or is this the text of the generated speech? Or sth else?


> The input of variation network comprises the prompt representation (P1, ..., PM ), noised reference representation (R1t , ..., PMt ), and diffusion step t

How exactly? P and R are concatenated sequences? But the diffusion process only runs on R?






Clarification on text attributes for text prompt generalization: There is only gender (via the existing SLU model), pitch, volume, and speed, nothing else? I would expect some more attributes, e.g. different emotion categories, etc.


Section 5.1, table 1: I don't exactly understand what is measured under what conditions on what data. So, each TTS model (VoiceGen vs the others) generates some speech, then some attributes are given to produce some prompt, and then, for the given fixed SLU models and digital signal processing tools, the accuracy is measured? That uses the generated prompts via LLM as mentioned before? All of them, so 20K text prompts? How many classes are there for each of the attributes?


Text prompt dataset: So this is released? Where? I did not find it.

> WavLM-TDNN model (Chen et al., 2022a) to assess the similarity of two speech

Is this a pretrained model which you just take? Or where is the model from?

I don't exactly understand Section 5.1, table 3. This is always with the same phoneme sequence as input to the TTS model? But what is the text context here? Why are the results so different between PromptTTS, InstructTTS and VoiceGen? Are they really comparable? Do they have the same TTS backbone? Does the TTS backbone use diffusion in all cases?

When you add variability through the variation network, how well are the properties of the text prompt for the speech style actually preserved? I guess this is table 1? But how much variance do you get when you add sampling results of the variation network?

---

> ### Author Response · Authors · 2023-11-19
> **Response to Reviewer Ghqv (1/2)**
>
> We sincerely appreciate your efforts in reviewing our paper and providing us with valuable and constructive feedback. Your comments have greatly benefited our work. We have addressed your questions below.
>
> **Q1: About the speech prompts and text prompts**
>
> Similar to your comments, a text prompt is a description of the desired output voice, such as “Please generate a voice of a boy shouting out”.
>
> For the speech prompt, it is **not** the speech version of text prompt. The speech prompt can be a speech of target style (maybe a speech of a famous person). The speech prompt based TTS model can learn from the speech prompt to generate speech in the same style (or voice) as the speech prompt. However, identifying an appropriate speech prompt can be a time-consuming task and may raise ethical concerns. Therefore, text description can be a flexiable and safe way for voice generation.
>
> **Q2: About the input of variation network**
>
> To provide a more precise analysis, we will discuss the problem with numerical details. The variation network we employed is a Transformer-based model, with the prompt representation, noised reference representation, and reference representation all set to a length of 8. The prompt representation and noised reference representation both have a shape of [batchsize, 8, hiddensize], which are concatenated to form a shape of [batchsize, 8+8, hiddensize]. This concatenated representation is then fed into the Transformer model, with the output also having a shape of [batchsize, 8+8, hiddensize]. The latter 8 outputs, corresponding to the noised reference representation, with a shape of [batchsize, 8, hiddensize], are used as the predicted results, along with the reference representation, for loss calculation.
>
> It should be noted that although we only consider the latter 8 outputs, the prompt representation can also influence the output through self-attention. Our approach of using a single model for both prompt and noised reference representations was inspired by the implementation in DALLE-2 (https://github.com/lucidrains/DALLE2-pytorch).
>
> **Q3: About the clarification on text attribute and setting**
>
> In our submission, all results presented in Table 1 are based on a fair comparison of text data, with all models utilizing LLM-written data. Additionally, the SLU and DSP models are fixed for metric calculation. We also conducted experiments on all models using text data from PromptTTS, which has lower quality. The results of these experiments can be found in Q1 in [Response to Reviewer g23M](https://openreview.net/forum?id=NsCXDyv2Bn&noteId=j7nT1dF7U1) and the conclusion is similar with that in Table 1 of the submission.
>
> Regarding the classes, we used two classes for gender (male and female) and three classes (high, normal, low) for other attributes such as pitch, speed, and volume. We plan to continue our work by adding more attributes, such as emotion, in our future research.
>
> **Q4: About the code and dataset**
>
> We have updated the supplementary material to include the code for VoiceGen and the prompt generation pipeline, along with the generated dataset. It is worth noting that this pipeline can be utilized to generate prompts for other attributes by providing the necessary attributes, classes, and ChatGPT keys (further details can be found in the README file located in the data_pipeline folder).
>
> **Q5: About the WavLM-TDNN model**
>
> We use the official model in https://github.com/microsoft/UniSpeech/blob/main/downstreams/speaker_verification/README.md.

---

> > ### Author Response · Authors · 2023-11-19
> > **Response to Reviewer Ghqv (2/2)**
> >
> > **Q6: About the clarification on Table 3 and Table 1**
> >
> > Regarding Table 3, as you have pointed out, the phoneme sequence is always the same for the "Text Prompt", "Variation Network", and "TTS Backbone" columns. And all the models utilize the same diffusion-based TTS backbone.
> >
> > The "Text Content" column is included for the sake of completeness, as some readers may argue that an extremely overfit TTS system can synthesize different voices or styles with different text content. However, this is not an issue for any of the baseline systems or VoiceGen.
> >
> > It is important to note that Table 3 (as well as Table 6) primarily demonstrates two key findings: 1) the variation network is capable of sampling different voices that correspond to the intention of a text prompt, and 2) the variability in voice is primarily controlled by the sampling results of the variation network, rather than other factors such as text prompt with the same meaning, text content to synthesize, or TTS backbone.
> >
> > Regarding Table 1, as you have noted, the synthesized results can occasionally be inconsistent with the text prompts. Increasing the number of samples can increase the probability of obtaining inconsistent results, although this can be alleviated through engineering techniques (such as utilizing a selection model to choose a result that is more consistent with the prompt, as leveraged in DALLE-2 [3]).
> >
> > **Q7: About the further ablations**
> >
> > We further conduct two ablations on the TTS backbone and the length of prompt and reference representations.
> >
> > Besides the TTS backbone used in the submission (i.e., latent diffusion in NaturalSpeech 2 [1]), we further apply our method (as well as the baseline methods) on another TTS backbone based on the token prediction of codec results (as proposed in SoundStorm [2]). The results are as follows:
> >
> > |  Model   | Gender | Speed | Volume | Pitch | Mean |
> > |  ----  | ----  |  ----  | ----  |  ----  | ----  |
> > | PromptTTS  | 97.47 | 89.96 | 92.41 | 85.06 | 91.23 |
> > | InstructTTS  | 96.55 | 89.66 | 92.18 | 84.14 | 90.63 |
> > | VoiceGen  | 98.92 | 92.18 | 93.95 | 90.73 | 93.95 |
> >
> > From the result, we observe that VoiceGen consistently outperform baseline methods in the condition that TTS backbone in SoundStorm [2] is leveraged.
> >
> > Moreover, we conduct experiments on a unique hyper-parameter in VoiceGen, i.e., the length of prompt and reference representations. The results are shown as follows:
> >
> > |  Representation Length   | Gender | Speed | Volume | Pitch | Mean |
> > |  ----  | ----  |  ----  | ----  |  ----  | ----  |
> > | 4  | 98.54 | 92.49 | 93.26 | 88.66 | 93.24 |
> > | 8 (Default Choice)  | 98.23 | 92.64 | 92.56 | 89.89 | 93.33 |
> > | 16  | 98.77 | 92.18 | 93.95 | 90.27 | 93.79 |
> > | 32  | 98.39 | 91.80 | 91.19 | 90.27 | 93.09 |
> >
> > From the results, we observe the length of representation is not a sensitive hyper-parameter in terms of performance.
> >
> > **Finally, we would like to express our gratitude again to the reviewer for their time and effort in reviewing our paper. Please do not hesitate to let us know if you have any further concerns or comments. We would be happy to address them.**
> >
> >
> > **Reference**
> >
> > [1] NaturalSpeech 2: Latent Diffusion Models are Natural and Zero-Shot Speech and Singing Synthesizers, Kai Shen, et al, 2023.
> >
> > [2] SoundStorm: Efficient Parallel Audio Generation, Zalán Borsos et al, 2023.
> >
> > [3] Hierarchical Text-Conditional Image Generation with CLIP Latents, Ramesh, A, et al, 2022.

---

### Official Review · Reviewer_g23M · 2023-10-31

**Soundness:** 3 good
**Presentation:** 3 good
**Contribution:** 3 good
**Rating:** 6
**Confidence:** 3

**Summary:**

This paper studies the problem of text-based voice creation for text-to-speech synthesis (TTS).
Prior work on zero-shot TTS often relies on using reference voice samples of the target speaker (YourTTS) or target audio style (including both speaker and prosody, such as VALL-E) to prompt the model to generate the desired voice.
However, the authors argue that such prompts may not always be available, and this paradigm is less user friendly. To address it, authors present a model to enable creation of voices through providing descriptions like “a man with a normal voice”, similar to the setup in InstructTTS and PromptTTS.
The contribution of the proposed method is two-fold.
First, the authors tackle the one-to-many problem between text description and voice, where the same description, such as “a low pitched female voice”, can be mapped to many different voices. The authors adopt a variation network to sample the reference speech style embeddings given a text description prompt.
Second, the authors presented a pipeline to automatically create text prompts to address the data scarcity issue for descriptive texts for speech. The authors consider controlling four aspects of speech: gender, speed, volume, and pitch.
In addition, the authors present a face2voice application replacing text description with facial image.

**Strengths:**

1. This paper studies an interesting problem which enables creation of voices through text descriptions. This line of research has great potential of making speech generation more customizable.

2. The authors present a systematic pipeline to produce text describing four aspects of speech, addressing the data scarcity problem. Ablation studying Table 7 shows the benefit of the step-by-step generation process.

3. The variation model tackles the one-to-many problem. The author verified that when changing variation networks introduce speaker variation in Tabel 3.

**Weaknesses:**

1. I am not certain if the proposed model and the baseline models are trained on the same data, and hence I cannot draw conclusions that whether the proposed model outperforms the baselines because of the additional LLM generated data or because of the introduction of the variational network to address the one-to-many problem. It would be good to show how well the baseline performs with and without LLM-augmented text prmopts

2. Given that the number of attribute combinations is rather small (2 x 3 x 3 x 3 = 54), I am suspicious about how useful it is to increase the number of text prompts. The author could have conducted ablation studies comparing using only the PromptTTS prompts vs those + x LLM-augmented prompts

3. The authors did not give sufficient background on InstructTTS. That model also deploys a diffusion model and in principle would be capable of modeling one-to-many mapping. Why does the VoiceGen model perform better than InstructTTS?

4. Given that the conditional attributes are all categorical, it can be conditioned with a lookup table for each attribute straightforwardly. How does that method compare with the proposed method? The paper does not really showcase the strength of free-form text description for conditioning attributes that are hard to categorize.

5. For step 1 when generating phrases, the authors show in Table 5 that male can be mapped to man/boy/guy/dude/gentlement. However, one would expect quite different voices between boy and man. This shows the weakness of such pipeline where text descriptions are created from underspecified labels.

**Questions:**

See the questions in the Weakness section

---

> ### Author Response · Authors · 2023-11-19
> **Response to Reviewer g23M**
>
> We sincerely appreciate your efforts in reviewing our paper and providing us with valuable and constructive feedback. Your comments have greatly benefited our work. We have addressed your questions below.
>
> **Q1: About the text data (Weakness 1 and 2)**
>
> Since the ablation study on text data shows the superiority of LLM-written data over the prompts in PromptTTS (as reported in Table 4 of the submission), the results in Table 1 of the submission are conducted when all the models using the LLM-written data. Thus, the results are a fair comparison in terms of text data.
>
> To strengthen the conclusions of our paper, we conducted additional experiments on all models (i.e., PromptTTS, InstructTTS, and VoiceGen) using the text data in PromptTTS. The results are in the following table:
>
> |  Model   | Text Dataset  | Gender | Speed | Volume | Pitch | Mean |
> |  ----  | ----  |  ----  | ----  |  ----  | ----  |  ----  |
> | PromptTTS  | Prompts in PromptTTS | 98.93 | 87.43 | 89.35 | 85.44 | 90.29 |
> | PromptTTS  | LLM-written | 98.01 | 89.66 | 92.49 | 85.95 | 91.54 |
> |  ----  | ----  |  ----  | ----  |  ----  | ----  |  ----  |
> | InstructTTS  | Prompts in PromptTTS | 96.55 | 86.13 | 88.43 | 85.52 | 89.16 |
> | InstructTTS  | LLM-written | 97.24 | 90.57 | 91.26 | 86.82 | 91.47 |
> |  ----  | ----  |  ----  | ----  |  ----  | ----  |  ----  |
> | VoiceGen  | Prompts in PromptTTS | 98.77 | 90.80 | 90.57 | 89.58 | 92.43 |
> | VoiceGen  | LLM-written | 98.23 | 92.64 | 92.56 | 89.89 | 93.33 |
>
> From the Table, we observe that 1) VoiceGen outperforms baseline methods on both text prompt datasets; 2) using LLM-written data consistently improves the quality of prompt-based TTS on all models, compared to using the prompts in PromptTTS.
>
> **Q2: About the diffusion in InstructTTS**
>
> We apologize for not discussing InstructTTS in our paper. InstructTTS addresses the one-to-many challenge by leveraging speaker ID (once the speaker id is provided, the variability on voice becomes smaller). However, this approach has two limitations. First, it restricts the application of InstructTTS to the speakers in the training dataset. Second, it prevents the voice generation model from modeling timbre, which is an important attribute of voice generation. In contrast, VoiceGen can create different voices, including variability in timbre, for the same text prompt.
>
> In addition, InstructTTS has also attempted to address the one-to-many problem by regularizing the reference encoder and text encoder outputs to converge in the embedding space using L2 loss. However, this approach is limited in its ability to resolve the one-to-many problem, as the output of the reference encoder contains much more information than that of the text encoder.
>
> In contrast, we approach the one-to-many problem as a generation task and utilize generative models (specifically, diffusion models) to generate the output of the reference encoder based on the output of the text encoder. VoiceGen is the first framework capable of generating and fixing virtual speakers. This is a significant contribution to both future research and the real-world application of prompt-based TTS systems.
>
> **Q3: About the categorical attributes (Weakness 4 and Weakness 5)**
>
> Firstly, regarding the effectiveness of text descriptions, we found that it outperforms the categorical lookup table method when controlling the voice with text description, as reported in PromptTTS[1], and we also confirmed this in our preliminary study.
>
> Secondly, for conditioning attributes that are difficult to categorize, we believe that the extension of Face2Voice to our model partially demonstrates its ability to generate voice based on unspecific information.
>
> Thirdly, we acknowledge that using more specific labels and descriptions for hard-to-categorize attributes would be beneficial. We plan to address this in our future work. We sincerely hope that our paper's contribution in presenting a general framework for modeling voice variability and text prompt writing can be recognized.
>
>
>
> **Finally, we would like to express our gratitude again to the reviewer for their time and effort in reviewing our paper. Please do not hesitate to let us know if you have any further concerns or comments. We would be happy to address them.**
>
> **Reference**
>
> [1] PromptTTS: Controllable text-to-speech with text descriptions, Zhifang Guo, et al, 2023.

---

> > ### Author Response · Authors · 2023-11-22
> > **Looking forward to further discussion**
> >
> > Dear Reviewer g23M,
> >
> > As the deadline for discussion is approaching, we would like to inquire that whether we have effectively addressed the questions raised in the initial review. If you have any further question, we would be very happy to reply.
> >
> > Thank you again for your time and effort in reviewing.
> >
> > Best, Authors

---

> > > ### Comment · Reviewer_g23M · 2023-11-23
> > > **Thank you for the response**
> > >
> > > 1. The author addressed my question. I would caution the authors against stating “using LLM-written data consistently improves the quality”. Gender accuracy degrades for PromptTTS and VoiceGen when using LLM-written.
> > > 2. I thank the author for the explanation. I see that the L2 loss in the embedding space do encourage reference encoder and text encoder to produce same embeddings. However, conditioned on the embedding (and other embeddings like content and speaker, as illustrated in Fig 5), InstructTTS still has the ability to generate diverse output, given the denoising Transformer models a stochastic process that could produce different output when initial noise (x^T) is different. Hence, in terms of modeling capability I do not see InstructTTS would have limitation of underfitting the one-to-many distribution. Nevertheless, I do agree with the authors that requiring speaker ID imposes another limitation on InstructTTS.
> > > 3. Thank you for sharing the empirical finding that conditioning on text description outperforms conditioning on categorical lookup table. However, from information theoretical perspective, I do not see how this could happen, given all the captions are derived from a finite set of values for each category (e.g., male/female for gender) and hence text description does not provide more information than the categorical attributes (and is even a noisy version of that). Could the other provide hypothesis on why this would happen? I do see the value for Face2Voice which I believe highlight the strength of the approach (when attributes are hard to categorize)
> > >
> > > Overall I think the authors have addressed many of my questions, while I still have doubts on some of the results (which are minor and could be noise from empirical studies). I have raised my score to reflect this.

---

> ### Author Response · Authors · 2023-11-23
> **Thanks for your comments**
>
> Dear Reviewer g23M,
>
> We are very delightful to see that most of the issues are addressed! Thanks for your further comments!
>
> For the comment 1, thank you for your suggestion and we have changed the statement (in Appendix C of the revised paper) to avoid over-claiming.
>
> For the comment 2, we agree that the L2 loss in InstructTTS is effective with more condition information such as speaker ID embedding or other embeddings. Considering the scenario of creating and fixing different virtual speakers or of limited conditional information, VoiceGen is a more general framework for voice generation compared with baseline methods since it can predict the missing information about voice variability.
>
> For the comment 3, we understand your concern from information theoretical perspective. However, the input of the prompt-based TTS system is the text prompt instead of category label. So if we want to use the category-conditioned model, we need to recognize category from the text prompt, which is not perfect and can lead to cascaded error. If the category is (100%) accurately recognized, then the category based method can be quite competitive. Besides the above explanation, it is worth noted that prompt based TTS system has a potential on modeling hard-to-categorize attributes (which is shown in the Face2Voice and is our important future work).
>
> Thanks again for your comments and the time for reviewing our paper!
>
> Best, Authors

---

### Meta-Review · Area_Chair_W9pS · 2023-12-04

**Metareview:**

This paper studies text-to-speech synthesis (TTS) using text-based voice descriptions. Prior work on zero-shot TTS generally relies on reference samples of the target speaker voice (YourTTS) or target audio style (including both speaker and prosody, such as VALL-E) to allow the model to generate the desired voice. Such samples however may not always be available and authors present a model to enable creation of voices through providing descriptions like “a man with a normal voice”, similar to the setup in InstructTTS and PromptTTS. The contribution of the proposed method is two-fold. First, the authors tackle the one-to-many problem between text description and voice, where the same description, such as “a low pitched female voice”, can be mapped to many different voices. The authors adopt a variation network to sample the reference speech style embeddings given a text description prompt. Second, the authors presented a pipeline to automatically create text prompts to address the data scarcity issue for descriptive texts for speech. The authors consider controlling four aspects of speech: gender, speed, volume, and pitch. In addition, the authors present a face2voice application replacing text description with facial image, e.g. synthesizing a voice that matches a target user image, surpassing the corresponding sota.

Strengths

- This paper studies an interesting problem which enables creation of voices through text descriptions. This line of research has great potential of making speech generation more customizable.
- The authors present a systematic pipeline to produce text describing four aspects of speech, addressing the data scarcity problem. Ablation studying Table 7 shows the benefit of the step-by-step generation process.
- The image-based variant outperforms existing works based on image-based TTS voice generation and opens up further avenues of analysis

Weaknesses

- The paper is a bit unclear on several details, but authors are releasing code and data, plus will improve the paper.
- All the experiments basically just show that the proposed model works well and solves the outlined problem. However, there is almost no analysis or ablation studies, etc.

**Justification For Why Not Higher Score:**

The paper leaves out ablation studies, although some have been provided in author response.
Reviewers feel that the paper is quite empirical, some more formal description of the problem (and a more generalizable approach) may help to turn this into a spotlight or oral paper.

**Justification For Why Not Lower Score:**

The paper is overall well written and solves a novel, interesting and relevant problem that deserves to be presented.

---

### Decision · Program_Chairs · 2024-01-16

Accept (poster)